# Individual and community-level factors of treatment-seeking behaviour among caregivers with febrile children in Ethiopia: A multilevel analysis

Bikis Liyew[1]*, Gebrekidan Ewnetu Tarekegn[2], Tilahun Kassew[3], Netsanet Tsegaye[1], Marye Getnet Asfaw[1], Ambaye Dejen Tilahun[1], Ayalew Zewdie Tadesse[4], Tesfa Sewunet Alamneh[2]

1 Department of Emergency and Critical Care Nursing, School of Nursing, College of Medicine and Health Sciences, University of Gondar, Gondar, Ethiopia, 2 Department of Biostatistics and Epidemiology, Institute of Public Health, University of Gondar, Gondar, Ethiopia, 3 Department of Psychiatry, College of Medicine and Health Sciences, University of Gondar, Gondar, Ethiopia, 4 Department of Emergency Medicine and Critical Care, St Paul's Hospital Millennium Medical College, Addis Ababa, Ethiopia

* biksliyew16@gmail.com

**Data Availability Statement:** All relevant data are within the paper and its Supporting information files.

## Abstract

### Background

Early diagnosis and treatment of childhood fever are essential for controlling disease progression and death. However, the Treatment-seeking behaviour of caregivers is still a significant challenge in rural parts of the African region. This study aimed to assess individual and community-level factors associated with treatment-seeking behaviours among caregivers of febrile under-five age children in Ethiopia.

### Method

The recent Ethiopian Demographic and Health Survey data (EDHS 2016) was used for the study. The survey collected information among 1,354 under-five children who had a fever within two weeks before the survey. The data were extracted, cleaned, and recoded using STATA version 14. Multilevel logistic regressions were used to determine the magnitude and associated factors of treatment-seeking behaviour among caregivers with febrile children in Ethiopia. Four models were built to estimate both fixed and random effects of individual and community-level factors between cluster variations on treatment-seeking behaviour. The Adjusted Odds Ratios with 95% Confidence Intervals (CI) of the best-fitted model were reported at $p < 0.05$.

### Result

This study revealed that 491 (36.26%) caregivers seek treatment for their febrile children. Living in metropolitan and small peripheral regions, delivery at health institutions, being poorer, middle and richer wealth quintiles, having a child with diarrhoea, cough, short rapid

**Funding:** The author(s) received no specific funding for this work.

**Competing interests:** The authors have declared that no competing interests exist.

breathing, and wasting were positively associated with treatment-seeking behaviour of caregivers.

## Conclusion

The caregivers had poor treatment-seeking behaviour for their febrile children in Ethiopia. Health education programmers should emphasise the importance of seeking early treatment, taking action on childhood febrile illness signs.

## Introduction

Treatment-seeking behaviour is the sequence of actions and integral part of a person's, a family's, or a community's identity that patients and caregivers take to solve their problem [1–3]. Febrile illnesses are complicated to diagnose and manage clinically because of the large variety of fever causing infections with similar clinical manifestations [4]. In developing countries, the most common cause of referral is an acute febrile illness [5]. Fever is most prevalent in African countries, such as Ethiopia and Tanzania [6, 7]. Malaria and pneumonia are the leading causes of morbidity and mortality in under-five children in these countries [8, 9]. In the study conducted in the rural communities of Somalia, the majority of fevers (84.4%) were associated with other symptoms, including cough, running nose, and sore throat, and only 37.5% of fever cases were managed at a formal health care facility [10, 11].

Moreover, fever treatment within 48 hours with effective anti-malarial is used as the milestone parameter in national sample surveys to measure the success of malaria case management policies [10]. A study conducted in Zambia showed that seeking early and appropriate treatment was suboptimal [12]. In malaria-endemic areas, fever has been used as a proxy for malaria even though the cause could be different [13, 14]. World Health Organization (WHO) 2016 report showed that approximately 34% of households of caregivers in sub-Saharan Africa seek early treatment for their febrile children from health care professionals [15]. In many sub-Saharan Africa countries such as Malawi [16], Liberia [17], Tanzania [18], and Zambia [12], the magnitude of caregivers' early treatment-seeking behaviour for febrile children was 67.3%, 98.5%, 56.8%, and 27.0%, respectively. Enhancing treatment-seeking behaviour regarding fever in Ethiopia is imperative [19–21]. In Ethiopia, healthcare-seeking behaviour is poor, and only 35% of under-five children with fever receive appropriate treatment [22]. Even though malaria remains a major cause of fever, its incidence has been steadily alleviated since 2003 [23, 24].

Bloodstream infection, including bacterial zoonosis, is one of the challenges for healthcare providers with similar clinical manifestations among hospitalised patients [25]. WHO has proposed guidelines on the Integrated Management of Childhood Illness regarding appropriate supportive care at the health facility level for those febrile children requiring antibiotic treatment due to the inadequacy of health care facilities and skilled health care workers. Febrile children are treated through the informal sector, which greatly impacts the treatment of many childhood illnesses and contributes to morbidity and mortality [26]. Different works of literature reported that seeking fever treatment is influenced by caregivers' knowledge, educational status, age, and availability of health facilities [27–30]. In low-income countries, community-based agents are the first people to treat childhood fever [31–34]. However, there was a significant increase in treatment-seeking behaviour for febrile children at the community level after the introduction of health care providers advised in malaria control [35]. In Sub-Saharan Africa, diseases causing fever contribute a significant impact on health [4].

The study conducted in different African countries reported multiple predictors of treatment-seeking behaviour among caregivers. These predictors were expendable income, economic constraint, previous history of a febrile patient, inaccessibility to health services, and undiagnosed perceived cause of febrile symptoms always linked to malaria were major significant predictors of treatment-seeking behaviour among caregivers [36–39]. Although fever is identified as a clinical sign of patients at Emergency and ICU departments and is the reason for the rational use of antibiotics, three are no studies identifying the treatment-seeking behaviours of caregivers for febrile children in Ethiopia [40]. Lack of treatment-seeking behaviour for childhood febrile illness among caregivers of under-5 children is still a major concern [29, 31]. Individual related factors such as the child's and caregiver's related factors [41], media exposure [42], caregiver's perceived distance to a health facility (no problem or problem), antenatal care (ANC) visits [43], the use of postnatal care, place of delivery [44], community development index [44, 45] were predictors of treatment-seeking behaviour reported in the literature. However, predictors of treatment-seeking behaviour among caregivers and the extent of febrile illness in Ethiopian contexts is not known. Therefore, the current study aimed to determine the factors associated with treatment-seeking behaviour among caregivers with febrile under-five children in Ethiopia. Identifying caregivers' treatment-seeking behaviour will help to create awareness and to take prompt action.

## Methods and materials

### Study design, setting, and period

We have used the 2016 Ethiopian Demographic and Health Survey (DHS) dataset (EDHS 2016). In Ethiopia, there were nine regional states such as Tigray, Afar, Amhara, Oromia, Somali, Benishangul-Gumuz, Southern Nations Nationalities and People Region (SNNPR), Gambela and Harari, and two administrative cities (Addis Ababa and Dire-Dawa). Eighty-four percent (84%) of the population lives in rural areas. The EDHS has consisted of a sample of households obtained through a two-stage stratified sampling procedure. The survey used the Ethiopian Population and Housing Census carried out in 2016 by the Ethiopia Central Statistical Agency (CSA) as the sampling frame. In the first stage, the country was divided into 645 (202 in urban and 443 in rural areas) primary sampling units or Enumeration Areas (EAs) by using the probability proportional to the size allocation method. In the second stage, a household listing was obtained in all selected EAs as a sampling frame, and an equal probability systematic sampling technique was carried out to select 28 households per cluster using the household listing. All women aged 15–49 who were either permanent residents of the selected households or visitors who stayed in the household the night before the survey were eligible to be interviewed. The 2016 EDHS 84915 enumeration areas (EA$_S$) served as a sampling frame, 645 clusters were selected in the first stage, and from those clusters, 202 were urban, and 443 were from rural areas.

A total of 16,650 households were surveyed in the second stage. The 2016 EDHS interviewed a total of 15,683 women between the ages of 15–49 years. The data for the present study were extracted as follows: First, women who gave birth in the last five years were identified. Next, caregivers/mothers who had a child with fever were identified in the two weeks preceding the survey period. It is customary to get more than one under-five child per household. However, for data quality, if more than one under-five children per household, data were collected from children with the last (recent) birth. Finally, of the 10,417 under-five children, 10,006 were alive children, and a total of 1,354 (weighted = 1495) mothers/caregivers who had under-five children with fever were included for the analysis. A total of 1,354 caregivers/mothers with febrile children under five were used to analyse this study. Caregivers were defined as

mothers aged between 15–59 years with a child/child under age 5 years who responded to the survey [46]. Fever is an abnormally high body temperature, usually accompanied by shivering, headache, and restlessness. Fever indicates the presence of various illnesses such as malaria, pneumonia, an ear problem, the common cold, influenza, and other infections. Treatment of fever is a Children with fever for whom advice or treatment was sought. The Sample of this EDHS study was Children under age 5 with fever two weeks before the survey, and the detailed sampling procedure was presented in the full 2016 EDHS report [47].

## Variables of the study

The outcome variable for this study was treatment-seeking behaviour (Yes/ No), defined as whether or not a caregiver sought advice or treatment from a health facility for a living child under five who had a fever at any time in the two weeks preceding the survey. The advice or treatment was sought from a governmental or private health facility by a health care professional [44, 48–50]. Socio-demographic and other health-related variables were included as independent variables. The socio-demographic variables were maternal age (year), child age (month), region, sex of household head, sex of the child, child's twin status, marital status, maternal educational level, and maternal currently working status. Other health related independent variables were size of child at birth, mass media exposure, wealth index combined, had diarrhoea, had a cough, had short rapid breathing, antenatal care, postnatal care, stunting, wasting, had anaemia, place of delivery, parity, vaccination, vitamin A in last six months, community development index, distance to the health facility, place of residence, and covered by health insurance.

Regions were categorised into three categories; Amhara, Oromia, Tigray, and SNNP regions were categorised as a large central region; the three administrative cities: Harar, Addis Ababa, and Dire Dawa, were categorised as a metropolitan region and the others (Somali, Gambelia, Afar, and Benishangul Gumuz region) were categorised as a small peripheral region. The aggregate community level explanatory variable: the community development index was constructed by aggregating individual-level characteristics at the cluster level by using an improved/unimproved source of drinking water, improved /unimproved sanitation facility, presence of electricity (no/yes) categorised as low, moderate, and good. Wealth Index was assessed to measure the socioeconomic status of the households based on household assets (television, bicycle/car, size of agricultural land, a quantity of livestock), and dwelling characteristics (sources of drinking water, sanitation facilities, and materials used for constructing houses), and the scores were divided into five categories of wealth quintile (poorest, poorer, medium, richer, and richest). Under-five children whose height-for-age Z-score, weight-for-age Z-score, and weight for height Z-score are below minus two standard deviations (− 2 SD) from the reference population's median are considered stunted underweight, and wasted, respectively. Percentage of children under age 5 with fever, diarrhoea, cough, and short rapid breathing at any time in the two weeks preceding the survey were included as independent variables. Child size at birth is the percent distribution of live births in the five years preceding the survey by mother's estimate of baby's size at birth (very small, smaller than average, average or larger, don't know/missing) recoded into large, average, and small. The vaccination status of the child is confirmed by vaccination card or mother's report. In those children, age 12–23 months and children age 24–35 months who received specific vaccines at any time before the survey according to vaccination card or mother's report by appropriate age recoded into complete and incomplete vaccination [51].

## Data processing and analysis

The variables of the study were extracted, cleaned, and recoded using STATA version 14. To accommodate for the complex sampling design employed in the survey. Weighted data

analysis was employed. Data weights were computed using sampling weights readily provided in the dataset and post-stratification weights developed by the researchers based on the 2016 population size of the nine regions and two city administrations of the country [52]. Descriptive statistics were performed with weighted data to explain the background characteristics of the individuals and communities. Four models (the intercept only (null model), individual-level factors (model ii), community-level factors (model iii), individual and community-level factors (model iv)) were fitted in these two levels of logistic regression analysis. During the analysis, caregivers (Level 1) were nested within their communities (Level 2) to estimate fixed effects of the individual and community-level factors and random effects between-cluster variation on treatment-seeking behaviours. Model I was the intercept-only multilevel logistic regression model (null model), which only included the outcomes of "treatment-seeking behaviour" to assess community effects on the treatment-seeking behaviour of the caregivers. In model II, the outcome and individual-level variables were fitted, whereas, with Model III, the outcome and community-level variables were included. Model IV fitted both the individual- and community-level variables.

Explanatory variables with a p-value of < 0.2 in the bivariable multilevel logistic regression model were fitted into the multivariable multilevel logistic regression model. A measure of association was reported as Adjusted Odds Ratio (AOR) with 95% CIs by controlling the effect of other predictors. A p-value<0.05 was used to identify factors significantly associated with treatment-seeking behaviours. Measures of variation (random effects) were assessed using several indicators. Variation between clusters (EAs) were assessed by computing Intra-lass Correlation Coefficient (ICC), the median odds ratio (MOR), and the proportional change in variance (PCV). The null model was used as a reference to look at the relative contribution in explaining TSB. The ICC is the proportion of variance explained by the grouping structure in the population. Whereas, PCV measures the total variation attributed to individual and community level factors in the multilevel model compared to the null model [53]. The goodness-of-fit of each model was assessed using the Akaike information criterion (AIC), and the Bayesian information criterion (BIC) was used for model comparison. A model with a lower AIC and BIC is preferred over a larger AIC and BIC model, which means a lower value representing a closer model fit, log-likelihood, and deviance; with lower deviance (Model IV) was the best-fitted model. These four models were compared using deviance (-2LLR), and the model with the lowest deviance was selected as the best-fitted model for the data. The multicollinearity effect has been checked; this was done by using the mean of variation inflation factor (VIF) and tolerance value. Less than ten mean VIF values indicate the absence of extreme collinearity problems among the regression model's explanatory variables. None of the variables displayed multicollinearity problems (all VIF < 10, tolerance > 0.1).

## Ethical consideration

This study is a secondary data analysis from the DHS data, so it does not require ethical approval. For conducting this study, online registration and request for measure DHS were conducted. The dataset was downloaded from DHS online archive (http://www.dhsprogram.com) after getting approval to access the data.

## Results

### Socio-demographic and health-related characteristics

Overall, this study revealed that 491 (36.26%) (95% CI: 33.74, 38, 86) caregivers sought treatment for their febrile children. This result shows that there are still a substantial number of febrile children under the age of five years that are not taken to health care facilities.

In this study, a total of 1354 febrile children were included from 10,006 alive children. Of the 1354 children, 534 (35.75%) have had diarrhoea in the past two weeks preceding the survey. The majority of caregivers, 898 (60.08%), had no education, and 1480 (98.98%) were married. Besides, the majority of the caregivers were not working 1,030 (68.93%) and not covered with health insurance 1,439 (96.25%). Regarding the age of the caregivers, 371(27.40%) were between 25–29 years of age, and the mean age was 28.90(SD = 6.52) years. Regarding the age of children, 393(26.31%) were between 12–23 months of age, and the mean age was 28.5 (SD = 17.45) months (Table 1).

## Random effect analysis and model comparison

As shown in Table 2, in the null model, the ICC indicated that about 12.2% of the total variability of treatment-seeking behaviour was due to differences between clusters/EA, with the remaining unexplained 87.8% was attributable to individual differences. Besides, the median odds ratio in the first model, which implies the caregivers within a cluster of having a higher risk for treatment-seeking behaviour had a 1.54 times higher chance of having treatment-seeking behaviour as compared with the caregiver within a cluster of having lower risk if caregivers were selected randomly from two different enumeration areas. Regarding PCV, about 54.0% of the variability in treatment-seeking behaviour was explained by the full model. Besides, Model IV was selected as the best-fitted model (lowest deviance = 1,553.86).

## Factors associated with treatment-seeking behaviour

In the final model, region, place of delivery, wealth index, having a child with diarrhoea, cough, short rapid breathing, and wasting were significantly associated with treatment-seeking behaviour of caregivers who had under-five febrile children (p<0.05). The odds of treatment-seeking behaviour were 2.04 (AOR = 2.04, 95% CI:1.296, 3.226) and 1.69 (AOR = 1.69, 95% CI: 1.191, 2.406) times more likely in the metropolitan region and small peripheral region than the large central region, respectively. Those caregivers who had wasted children were 1.55 (AOR = 1.55, 95%CI: 1.022, 2.359) times more likely to seek treatment than their counterparts. The odds of treatment-seeking behaviour was 1.44 (AOR = 1.44, 95%CI;1.054, 1.977) times higher in caregivers who had children with a history of cough than caregivers with under-five children that had no history of cough. Caregivers with poorer wealth index quantile had 1.60 (AOR = 1.60, 95%CI: 1.046, 2.453)) times more likely to seek treatment than caregivers with the poorest wealth index quantile by considering the other predictor variable constant. As compared to the poorest wealth index quantile, the odds of having treatment-seeking behaviour was 1.60 (AOR = 1.60, 95%CI:1.046, 2.453),1.89 (AOR = 1.89, 95%CI:1.207, 2.971), and 2.37 (AOR = 2.37, 95%CI: 1.49, 3.747) times higher for poorer, middle, and richer wealth index quintiles, respectively. Regarding short rapid breathing in the last two weeks, the odds of treatment-seeking behaviour among caregivers with children with a history of short rapid breathing was 1.68 (AOR = 1.68, 95%CI:1.242,2.269) times higher than their counterparts. Caregivers having under-five children with a history of diarrhoea had 1.53 (AOR = 1.53, 95% CI:1.166, 2.009) times more likely to seek treatment than caregivers who had under-five children with no history of diarrhoea. Moreover, the odds of treatment-seeking behaviour were 1.44 (AOR = 1.44, 95% CI:1.027, 2.015) times more likely for caregivers delivered at the institution than caregivers delivered at home (Table 2).

## Discussion

This study reported that 36.26% (95% CI: 33.74, 38.86) of caregivers sought treatment for their febrile children. This finding was lower than previous studies conducted in Malawi [16],

**Table 1. Weighted socio-demographic and health-related characteristics of the study participants in Ethiopia, 2016(n = 1354; weighted sample = 1495).**

| Respondent's characteristic | Categories | Weighted Sample | |
|---|---|---|---|
| | | Frequency | Percentage |
| Maternal age in years | 15–19 | 43 | 2.85 |
| | 20–24 | 337 | 22.54 |
| | 25–29 | 432 | 28.91 |
| | 30–34 | 351 | 23.50 |
| | 35–39 | 215 | 14.35 |
| | 40–44 | 83 | 5.58 |
| | 45–49 | 34 | 2.27 |
| Childs age in months | 0–11 | 370 | 24.77 |
| | 12–23 | 394 | 26.31 |
| | 24–35 | 287 | 19.22 |
| | 36–47 | 228 | 15.26 |
| | 48–59 | 216 | 14.44 |
| Region | Large central region | 1,382 | 92.42 |
| | Metropolitan region | 43 | 2.89 |
| | Small peripheral region | 70 | 4.69 |
| Sex of household head | Male | 1284 | 85.91 |
| | Female | 211 | 14.09 |
| | Total | 1495 | 100 |
| Sex of child | Male | 768 | 51.40 |
| | Female | 727 | 48.60 |
| Child is twin | Single birth | 1448 | 96.88 |
| | 1st of multiple | 24 | 1.60 |
| | 2nd of multiple | 23 | 1.52 |
| Marital status | Unmarried | 15 | 1.02 |
| | Married | 1,480 | 98.98 |
| Maternal educational level | No education | 898 | 60.08 |
| | Primary | 481 | 32.20 |
| | Secondary | 79 | 5.24 |
| | Higher | 37 | 2.49 |
| Maternal currently working | No | 1,030 | 68.93 |
| | Yes | 465 | 31.07 |
| size of child at birth | Large | 472 | 31.58 |
| | Average | 545 | 36.44 |
| | Small | 478 | 31.98 |
| Mass media exposure | No | 915 | 61.20 |
| | Yes | 580 | 38.80 |
| Wealth index combined | Poorest | 313 | 20.94 |
| | Poorer | 317 | 21.19 |
| | Middle | 306 | 20.47 |
| | Richer | 322 | 21.53 |
| | Richest | 237 | 15.88 |
| Had diarrhea | No | 961 | 64.25 |
| | Yes | 534 | 35.75 |
| Had to cough | No | 443 | 29.60 |
| | Yes | 1,052 | 70.40 |

(*Continued*)

**Table 1.** (Continued)

| Respondent's characteristic | Categories | Weighted Sample | |
|---|---|---|---|
| | | Frequency | Percentage |
| Had short rapid breathing | No | 784 | 52.47 |
| | Yes | 711 | 47.53 |
| Antenatal care | no visit | 719 | 48.08 |
| | 1–3 visit | 363 | 24.30 |
| | 4+ | 413 | 27.63 |
| Postnatal care | No | 1,382 | 92.44 |
| | Yes | 113 | 7.56 |
| Underweight | Yes | 1,084 | 72.54 |
| | No | 411 | 27.46 |
| Stunting | Yes | 988 | 66.08 |
| | No | 507 | 33.92 |
| | Total | 1495 | 100 |
| Wasting | Yes | 1,332 | 89.08 |
| | No | 163 | 10.92 |
| Had anemia | Severe | 41 | 2.73 |
| | Moderate | 419 | 28.02 |
| | Mild | 372 | 24.90 |
| | Not anemic | 663 | 44.36 |
| Place of delivery | Home | 1,007 | 67.35 |
| | Institution | 488 | 32.65 |
| Parity | Prim parous | 266 | 17.78 |
| | Multi parous | 618 | 41.31 |
| | Grand multipara | 611 | 40.91 |
| Vaccination | Incomplete | 1,199 | 80.18 |
| | Complete | 296 | 19.82 |
| vitamin A in last 6 months | No | 826 | 55.28 |
| | Yes | 669 | 44.72 |
| Community development index | Low | 288 | 19.27 |
| | Moderate | 682 | 45.61 |
| | Good | 525 | 35.12 |
| distance to a health facility | big problem | 795 | 53.15 |
| | not a big problem | 700 | 46.85 |
| Place of residence | Urban | 192 | 12.84 |
| | Rural | 1,303 | 87.16 |
| covered by health insurance | No | 1,439 | 96.25 |
| | Yes | 56 | 3.75 |

Liberia [17], and Tanzania [18] in that 67.3%, 98.5%, and 56.8%, respectively, of the caregivers, had treatment-seeking behaviour for their febrile child. The possible reason for the difference might be due to financial constraints, caregivers or mothers in Ethiopia visit traditional healers before they go to health care facilities, perception of caregivers regarding the illness, most of the caregivers/mothers in this study were rural residence, most of them also have no educational background, and, in Liberia, the information gathered about treatment-seeking behaviour was after a social and behaviour change campaign, and this might expose caregivers to malaria-related messages.

**Table 2. Multilevel analysis of factors associated with treatment-seeking behaviour among caregivers with febrile children in Ethiopia, 2016.**

| Respondent's characteristic | Categories | Null Model (I) | Model II AOR (95% AOR (95% CI) CI) | Model III AOR (95% AOR (95% CI) CI) | Model IV AOR (95% AOR (95% CI) CI) |
|---|---|---|---|---|---|
| | | | Individual-level factors | | |
| Maternal age in years | 15–19 | | 1 | | 1 |
| | 20–24 | | 0.69(0.348, 1.401) | | 0.74(0.365, 1.503) |
| | 25–29 | | 0.716(0.344, 1.491) | | 0.76(0.364, 1.608) |
| | 30–34 | | 0.45(0.204, 0.981)* | | 0.48(0.217, 1.069) |
| | 35–39 | | 0.54(0.234, 1.231) | | 0.57(0.247, 1.327) |
| | 40–44 | | 0.55(0.211, 1.415) | | 0.61(0.232, 1.595) |
| | 45–49 | | 0.44(0.108, 1.751) | | 0.54(0.131, 2.213) |
| Maternal educational level | No education | | 1 | | 1 |
| | Primary | | 1.31(0.951, 1.799) | | 1.35(0.977, 1.853) |
| | Secondary | | 1.26(0.724, 2.203) | | 1.10(0.626, 1.937) |
| | Higher | | 2.64(1.204, 5.795)* | | 2.09(0.941, 4.653) |
| Children's age in months | 0–11 | | 1 | | 1 |
| | 12–23 | | 0.83(0.564, 1.207) | | 0.80(0.549, 1.177) |
| | 24–35 | | 1.08(0.721, 1.635) | | 1.095(0.726, 1.649) |
| | 36–47 | | 0.99(0.631, 1.547) | | 0.941(0.600, 1.476) |
| | 48–59 | | 1.11(0.697, 1.779) | | 1.05(0.658, 1.691) |
| Child is twin | Single birth | | 1 | | 1 |
| | 1$^{st}$ of multiple | | 0.58(0.165, 2.031) | | 0.63(0.183, 2.186) |
| | 2$^{nd}$ of multiple | | 0.51(0.160, 1.643) | | 0.59(0.1836, 1.898) |
| Sex of child | Male | | 1 | | 1 |
| | Female | | 0.81(0.626, 1.044) | | 0.83(0.643, 1.075) |
| Had to cough | No | | | | 1 |
| | Yes | | 1.43(1.047, 1.952)* | | 1.44(1.054, 1.977)* |
| Had short, rapid breathing | No | | 1 | | 1 |
| | Yes | | 1.54(1.145, 2.079)* | | 1.679(1.242,2.269)* |
| ANC | no visit | | 1 | | 1 |
| | 1–3 visit | | 1.37(0.964, 1.945) | | 1.41(0.993, 2.009) |
| | 4+ | | 1.22(0.842, 1.769) | | 1.211(0.834, 1.757) |
| Wasting | No | | 1 | | 1 |
| | Yes | | 1.63(1.071, 2.469)* | | 1.55(1.022, 2.359)* |
| Stunting | No | | 1 | | 1 |
| | Yes | | 0.98(0.699, 1.361) | | 0.99(0.715, 1.396) |
| Underweight | No | | 1 | | 1 |
| | Yes | | 0.86(0.593, 1.235) | | 0.89(0.615, 1.285) |
| Maternal currently working | No | | 1 | | 1 |
| | Yes | | 1.12(0.842, 1.489) | | 1.08(0.810, 1.434) |
| size of child at birth | Large | | 1 | | 1 |
| | Average | | 1.01(0.744, 1.381) | | 1.03(0.752, 1.399) |
| | Small | | 0.89(0.638, 1.249) | | 0.91(0.649, 1.271) |
| Mass media exposure | No | | 1 | | 1 |
| | Yes | | 1.05(0.765, 1.428) | | 1.08(0.787, 1.471) |
| PNC | No | | 1 | | 1 |
| | Yes | | 1.05(0.676, 1.620) | | 1.05(0.679, 1.629) |
| Diarrhea | No | | 1 | | 1 |
| | Yes | | 1.55(1.183, 2.037)* | | 1.53(1.166, 2.009)* |

(*Continued*)

**Table 2.** (Continued)

| Respondent's characteristic | Categories | Null Model (I) | Model II AOR (95% CI) | Model III AOR (95% CI) | Model IV AOR (95% CI) |
|---|---|---|---|---|---|
| Delivery place | Home | | 1 | | 1 |
| | Institution | | 1.63(1.169, 2.260)* | | 1.44(1.027, 2.015)* |
| Parity | Prim parous | | 1 | | 1 |
| | Multi parous | | 1.26(0.845, 1.878) | | 1.22(0.814, 1.818) |
| | Grand multipara | | 1.33(0.795, 2.231) | | 1.33(0.793, 2.246) |
| wealth index combined | Poorest | | 1 | | 1 |
| | Poorer | | 1.32(0.879, 1.969) | | 1.60(1.046, 2.453)* |
| | Middle | | 1.578(1.032, 2.414)* | | 1.89(1.207,2.971)* |
| | Richer | | 1.93(1.241, 2.991)* | | 2.37(1.49, 3.747)* |
| | Richest | | 2.88(1.781, 4.670)* | | 1.83(0.995, 3.379) |
| Vaccination | Incomplete | | 1 | | 1 |
| | Complete | | 1.22(0.857, 1.736) | | 1.23(0.863, 1.755) |
| vitamin a in last 6 months | No | | 1 | | 1 |
| | Yes | | 0.99(0.758, 1.296) | | 0.99(0.756, 1.295) |
| Community-level factors | | | | | |
| Community development index | Low | | | 1 | 1 |
| | Moderate | | | 0.87(0.571, 1.341) | 0.96(0.612, 1.504) |
| | Good | | | 1.24(0.792, 1.928) | 1.29(0.811, 2.066) |
| Place of residence | Urban | | | 1 | 1 |
| | Rural | | | 0.32(0.214, 0.488)* | 0.61(0.336, 1.119) |
| covered by health insurance | No | | | 1 | 1 |
| | Yes | | | 1.70(0.885, 3.279) | 1.40(0.707, 2.788) |
| Region | Large central | | | 1 | 1 |
| | Metropolitan | | | 1.49(0.962, 2.319) | 2.04(1.296, 3.226)* |
| | Small peripheral | | | 1.07(0.786, 1.443) | 1.69(1.191, 2.406)* |
| Community-level variance | | 0.456(0.17) | 0.273(0.15) | 0.300(0.138) | 0.213(0.142) |
| ICC | | 0.122 | 0.076 | 0.083 | 0.061 |
| MOR | | 1.54 | 1.296 | 1.329 | 1.224 |
| PCV (%) | | Reference | 40.1 | 34.0 | 54.0 |
| Deviance | | 1759.78 | 1,582 | 1666.82 | 1,553.8599 |
| Log-likelihood | | -879.89 | -791.00 | -833.41 | -776.92995 |
| AIC | | 1763.776 | 1662.009 | 1682.823 | 1645.86 |
| BIC | | 1774.197 | 1870.442 | 1724.51 | 1885.558 |

Note: AOR = Adjusted Odds Ratio; CI = Confidence Interval; 1.00 = Reference Group * = P<0.05; *ICC = Intra-class Correlation Coefficient*: *MOR*: *Median Odds Ratio*: *PCV*: *Proportional Change in Variance*.

The magnitude of treatment-seeking behaviour was also lower than the study conducted in an urban area of Eastern Sudan, reported 40.4% and 91.1% of housewives seeking advice from health workers for mild fever and severe fever, respectively [54]. The possible justification might be due to the sample size, measurement tool, and approaches of the outcome variable.

This study used nationally representative data with 1,354 samples collected from caregivers who had under-five children with fibril illness. Whereas in eastern Sudan, in a single centred (Kassalacity) with 350 housewives only were interviewed, and the outcome variable was

categorised as available treatment options (ATP) concerning the intensity of fever (low and high). Therefore treatment-seeking behaviour varied from country to country due to evaluation of the symptoms, perceived treatment effects, the initial reaction of caretakers, and perceived intensity of fever, regardless of the illness [54]. A study conducted In Kenya reported that caregivers were started self-treatment at home and waited for some time to observe the progress of the illness [55]. This was likely to lead to a delay in the treatment of malaria and another unfortunate consequence.

Children who were reported to have had diarrhoea in the last two weeks were 1.53 times more likely to be taken for fever treatment than those who had no diarrhoea. This study was inconsistent with a study conducted in Malawi in which febrile children who had diarrhoea in the last two weeks were less likely to be brought for treatment than those who had no diarrhoea [16]. Compared to the poorest wealth index quantile, the odds of having treatment-seeking behaviour were 1.60, 1.89, and 2.37 times higher for the poorer, middle, and richer wealth index quantiles, respectively. This study was inconsistent with evidence from other sub-Saharan African countries, which reported that household wealth was not significantly related to caregivers delaying seeking care for children with fever [44, 56]. The possible justification might be that the diagnosis and treatment of fever in another country like Liberia are generally free of cost [57, 58]. Whereas in Ethiopia, the diagnosis and management are not free of charge, and due to the low economic status of the country, most caregivers are not affordable to pay for the treatment cost of febrile illness. There was no significant association between treatment-seeking behaviour and the caregiver's age and level of education, unlike a study done in Grand Gedeh County, Liberian, and Tanzania [18, 56, 59]. In this study, media exposure (radio, television, and newspaper) was not significantly associated with treatment-seeking behavior among care givers with febrile children. This study was consistent with a study conducted in Malawi [60]. However, it was inconsistent with a study conducted in Tanzania [18].

## Strength and limitations of the study

This study used nationally representative samples. Therefore, the results could be generalised to all Ethiopian caregivers/mothers. Fever was assessed by history that can lead to recall bias, but the 2-week recall period could help to reduce the bias. However, it is a cross-sectional study, did not signify causal attribution; the use of clusters or administratively defined boundaries could yield information bias for those caregivers from unfitted administrative communities; social desirability bias since self-reports on whether treatment was sought are prone to caregivers may attempt to present that they take good care of their children by, among others, seeking treatment whenever their children have a fever; and the study has not identified the influence of caregivers' knowledge, perceptions, and attitudes on different childhood illnesses that provoke fever such as malaria.

## Conclusion

This study revealed that the treatment-seeking behaviour of mothers'/care givers' in Ethiopia was low, even if a common childhood illness remains high. Therefore, in Ethiopia, improving treatment-seeking behaviour needs more attention and emphasis. Living in metropolitan and small peripheral regions, delivery at health institutions, being poorer, middle and richer wealth quintiles, having a child with diarrhoea, cough, short rapid breathing, and wasting were positively associated with treatment-seeking behaviour of caregivers. Therefore, both individual and community variables may prove fundamental to effect improved behaviours.

## Supporting information

**S1 File.**
(SAV)

**S2 File.**
(DTA)

**S1 Data.**
(XLS)

## Acknowledgments

We would like to acknowledge the Demographic and health survey (DHS) for providing permission to use the EDHS dataset for this analysis.

## Author Contributions

**Conceptualization:** Bikis Liyew.

**Data curation:** Bikis Liyew, Tilahun Kassew, Marye Getnet Asfaw, Ambaye Dejen Tilahun, Ayalew Zewdie Tadesse, Tesfa Sewunet Alamneh.

**Formal analysis:** Bikis Liyew, Gebrekidan Ewnetu Tarekegn, Tilahun Kassew, Netsanet Tsegaye, Marye Getnet Asfaw, Tesfa Sewunet Alamneh.

**Methodology:** Bikis Liyew, Gebrekidan Ewnetu Tarekegn, Tilahun Kassew, Netsanet Tsegaye, Marye Getnet Asfaw, Ambaye Dejen Tilahun, Ayalew Zewdie Tadesse, Tesfa Sewunet Alamneh.

**Software:** Bikis Liyew, Gebrekidan Ewnetu Tarekegn, Tilahun Kassew, Ambaye Dejen Tilahun, Tesfa Sewunet Alamneh.

**Supervision:** Netsanet Tsegaye, Marye Getnet Asfaw, Ambaye Dejen Tilahun, Ayalew Zewdie Tadesse.

**Writing – original draft:** Bikis Liyew, Netsanet Tsegaye.

**Writing – review & editing:** Ayalew Zewdie Tadesse, Tesfa Sewunet Alamneh.

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
