## [Decision Letter · Decision Letter 0]

2 Jun 2021

PONE-D-21-03591

Individual and community-level factors associated with treatment-seeking behavior among caregivers with febrile children in Ethiopia: A Multilevel analysis

PLOS ONE

Dear Dr. liyew,

Thank you for submitting your manuscript to PLOS ONE. After careful consideration, we feel that it has merit but does not fully meet PLOS ONE’s publication criteria as it currently stands. Therefore, we invite you to submit a revised version of the manuscript that addresses the points raised during the review process.

The paper needs substantial improvements in all parts in order to be readable. Please address reviewers feedback and revised the paper accordingly. 

We look forward to receiving your revised manuscript.

Kind regards,

Enamul Kabir

Academic Editor

PLOS ONE

Journal Requirements:

2. Please amend either the title on the online submission form (via Edit Submission) or the title in the manuscript so that they are identical.

3. Thank you for submitting the above manuscript to PLOS ONE. During our internal evaluation of the manuscript, we found significant text overlap between your submission and the following previously published works, some of which you are an author.

http://eprints.gla.ac.uk/228008/1/228008.pdf

https://journals.plos.org/plosone/article?id=10.1371%2Fjournal.pone.0202240

Please revise the manuscript to rephrase the duplicated text, cite your sources, and provide details as to how the current manuscript advances on previous work. Please note that further consideration is dependent on the submission of a manuscript that addresses these concerns about the overlap in text with published work.

Reviewers' comments:

Reviewer's Responses to Questions

**Comments to the Author**

1. Is the manuscript technically sound, and do the data support the conclusions?

Reviewer #1: Partly

Reviewer #2: Yes

2. Has the statistical analysis been performed appropriately and rigorously? 

Reviewer #1: Yes

Reviewer #2: Yes

3. Have the authors made all data underlying the findings in their manuscript fully available?

Reviewer #1: Yes

Reviewer #2: Yes

4. Is the manuscript presented in an intelligible fashion and written in standard English?

Reviewer #1: Yes

Reviewer #2: Yes

5. Review Comments to the Author

Reviewer #1: introduction

line 64 : authors should look at the sentence

line 65 : fever is a sign of illness and so what

lines 66 and 67 : is not in anyway relevant at this point of the introduction

lines 67 to 70: starting from diagnosing ......and ending with effects should be deleted

lines 72 and 73: authors should indicate if which people (household, individual or community) seek either early or late treatment

lines 74 to 76: Authors should at least make a case for SSA or Ethiopia as well

Discussion

lines 270 and 271 : Authors can not justify the statement, therefore, it should be rephrased or deleted

Reviewer #2: This is an important study looking at a significant issue in an understudied context. The authors employ appropriate methods for measuring associations between a variety of factors and treatment-seeking for childhood fever. With some major revisions particularly around the framing of the study (introduction), more information about the variables, sample construction, measurement and analysis (methods), and contextualized results (discussion) this paper can make a contribution to the literature around access and healthcare seeking behavior.

Comments by section

1) Abstract: cut off – probably due to word count issues. Would like to see a little more summary and contextualization instead of lists of results/AORs – especially because certain terms/concepts haven’t been defined yet when someone is reading abstract so can’t assume reader will be able to understand meaning of results at this point.

2) Introduction:

a. First paragraph: Awkward language especially in first sentence. What does the inclusion of evidence from Europe and US add to the study? US numbers not clear 4.4 -7.5% and 30%? Could be because of missing information/typos but it seems like Ethiopia has the highest treatment seeking rates (87.8%) for children of the countries listed so the following sentence about “enhancing treatment-seeking” is less compelling for Ethiopia (although obviously still important for the remaining 12.2% of cases). This could be made clearer and more compelling.

b. Overall flow of the introduction could be improved. It also seems to fall short of setting up what the study is really about. The authors provide compelling evidence that fever is linked to Malaria and that early treatment is important for morbidity and mortality, but the introduction is lacking logical flow and clear delineation between content in paragraphs. I would suggest that it be reorganized to set up study, not just in context of rates of fever and why fever matters, but more focus on treatment -seeking behavior.

i. For example, this section should introduce the predictors you seek to measure (generally) and help the reader understand why you are measuring certain individual and community level factors. What are the hypothesized relationships? Is there a conceptual framework or theoretical model that these are based on?

3) Methods and Materials:

a. Section 2.1 should state how child information was collected (e.g. by caregiver report). This comes up again in section 3.2 where it says “of the 10,006 alive children who responded.” I’m assuming that the babies and children are not the ones responding, but this should be very explicit. The total children (10,006) is not mentioned at all in this section. It should be clear where the subsample of 1,354 is drawn from. Also, it seems like caregivers who are not the mother are excluded (grandmothers, aunts, sisters etc.) but it should be explicit that the respondents are the mother to the child for whom the data is reported (not just the caregiver who is a mother).

i. What is the distribution of the selected subsample (1,354 children with fever)? We know about EDHS, but does this group have the same characteristics or are they concentrated in certain regional states etc.? This comes up later in discussion section – is this subsample representative? Perhaps as an appendix, it would be helpful to have a comparison of these children and caregiver characteristics compared to the whole sample.

b. Section 2.2 lists “private sector” “market” as a place to seek care. What is meant by this location?

c. Section 2.2 – would suggest adding another section on analysis and separating from the variables/measures

d. Section 2.2: The Independent variables listed do not align with what is included in the tables (9+ variables in tables not included in this section). Need more information on the variables. Would like to understand more about why certain predictors were included. What are the hypothesized relationships? Do these come from the literature? Are they context specific or more globally applicable factors? How were specific indices constructed (not just what variables were included in them) and are these based on particular measures from a source (WHO? UNICEF? USAID? Article?) Not all of this needs to go in the methods section, some could be in the introduction, but overall, more information is needed about what is being measuring and why.

e. Section 2.2: need to include information on the clusters (“communities”) how are they constructed. Why are they used instead of other levels?

f. How were the levels determined? Fever incidents are nested within children who are nested within caregivers, who are nested in families, who are nested in households, who are nested in communities, cities, states, regions, rural /urban etc. How were certain levels either eliminated (e.g. by selecting only 1 fever incident per child, only 1 child per caregiver, only 1 caregiver per household?) or otherwise accounted for? Of particular importance here, given the variables in your model, is children within caregiver and caregivers within a household.

g. Why did you use a cutoff of <=.26 for variable selection? This goes along with more information about the specific variables and why they were included, what they tell us about the research question.

h. How are missing data handled?

4) Results

a. 3.1 Background Characteristics: Major typo in first sentence that has serious implications if not fixed – “1006 alive children “Further reading showed this should be “10,006” . Another typo in that it references “unmarried as 1,479 (98.98%)” per table this is “married.” Also a minor point on language – “majority” typically means >50% so wouldn’t be used as it is in this paragraph to refer to 27.4% and 26.31%.

b. Table 1: Would be good to include totals so a reader can quickly see the sample size/total frequencies (and assess if they are consistent). As stated before – more information on the variables is needed. There are several variables here that were never mentioned before appearing in the table. While some are more obvious (sex of child), others are somewhat unclear (sex of household?, had diarrhea, had to cough, had anemia) – how were these operationalized, who is being measured, what is the time frame? Why do they matter? Obviously, not all of this information goes in the table, but some could be included here, whereas other info can be in the introduction and methods sections.

c. 3.3 Random effect…: More information about the clusters needs to be included in the methods section before reporting on the results by clusters. It is not clear how the clusters are constructed – what is the definition of “community.” How many communities are there? Is there sufficient power for the number of clusters and units within each cluster? The last sentence of this paragraph states that the Model IV was selected because it had the lowest deviance – but the number reported from Table 2 is different than the actual number listed in Table 2 for that model. Which is it?

d. 3.4 Factors associated…: Why state that variables had a statistically significant association at the level of <.20 when in previous sections (methods) you said you were using a cutoff of <0.05 (which is standard) ? In multiple sections/tables the coughing variable is not clearly explained. The text sometimes states that the children were coughing, sometimes the caregiver, sometimes the caregivers “had to cough febrile children.” This should be clear and I believe in the DHS the variable is measuring whether the children had a cough. Same thing applies to diarrhea and rapid breathing. It should be very clear who had the diarrhea and when (same time as fever?) It is assumed this is the child and simultaneous to the fever, but the text/tables are missing important descriptions or stating it in confusing/inconsistent ways (e.g. line 257 “among caregivers that were having short, rapid breathing was 1.68 times…” line 258 “mothers having diarrhea…” line 259 “caregivers who have no diarrhea”)

e. Table 2: needs proper formatting such as consistent column width.

5) Discussion/limitations/conclusion:

a. The Malawi numbers listed are inconsistent with what is in the introduction (49.9% vs. 67.3%) assuming these studies are measuring different things but this should be clear.

b. Discussion of sample size seems to be more about the sample construction, not the size? Unless there were issues with too few observations for the multilevel approach. This should be specifically addressed in methods and limitations section. (size, distribution, representativeness).

c. Would like to see more discussion about the specific predictors not just how they compare to prior studies, but also what they mean in context. Did the hypothesized relationships/effects play out? How do the different predictors relate to eachother (or not).

d. It is not clear that this can be generalized to all Ethiopian caregivers/mothers (also those are different groups) because we don’t know if this small subsample drawn from a representative sample is representative. Are mothers (or caregivers) of children under 5 who had a fever in the last 2 weeks prior to the survey representative of all mothers? Would like to see more in the limitations sections about any potential issues in the methodological and conceptual approaches.

e. Conclusion the mention of diarrheal disease derail treatment-seeking behavior is confusing and doesn’t seem to be in line with results.

f. Expected to see these sections linked back to the introduction more in terms of the focus on malaria, implications for the children etc.

g. The contribution of this study (which I believe in) isn’t coming through clearly . Being very precise about what was learned, what it means, and how these results can be used/interpreted is key.

6) General: Should be copy-edited for typos, awkward language, grammar issues.

6. PLOS authors have the option to publish the peer review history of their article (what does this mean?). If published, this will include your full peer review and any attached files.

Reviewer #1: No

Reviewer #2: No

---

## [Author Response · Author response to Decision Letter 0]

20 Jul 2021

Authors’ Response for Reviewers’ Comments

Manuscript ID: PONE-D-21-03591

Title: Individual and community-level factors associated with treatment-seeking behavior among caregivers with febrile children in Ethiopia: A Multilevel analysis

Dear editor(s) and reviewers

First for all the authors would like to thank the editor(s) and reviewers for your precious time, thoughtful comments and constructive suggestions, which help to improve the quality of this manuscript. We have responded to each critique/ comment and believe that the manuscript is much improved with the changes we made as suggested by the editor and reviewers. The corresponding changes and refinements made in the revised manuscript are summarized in our response below.

Response=> Authors response for editor and/ reviewers comments 

Reviewer #1

Evaluation (review comments for the authors)

Introduction

line 64 : authors should look at the sentence

Response: We thank the reviewer for raising this important point. We rewrote the introduction for greater focus. This section has been edited for conciseness by presenting the crucial points of what is known and not known, and why we carried out the study.

Q: line 65: fever is a sign of illness and so what

Response: After your suggestions, we have revised again the document. We have also noted repetitive sentences, contextual spelling and sentence structure problems. Then we have re-arranged and re-edited the vague sentences. We have provided the document for professional language copy editor expert to advanced language edition. Hence clarity problems are resolved and highlighted by red color in the revised manuscript.

Q: lines 66 and 67: is not in any way relevant at this point of the introduction

Response: This comment is crucial for this manuscript which was our gap. Now we have reviewed works of literature regarding the guidelines of physical restraint use in different countries including Ethiopia on the targeted and other population. Information regarding the issue and the rationale is provided in the introduction section of the revised manuscript and highlighted by red color

Q: lines 67 to 70: starting from diagnosing ......and ending with effects should be deleted

Response: thank you for your suggestion. The suggested correction has been made and highlighted in track change.

Q: lines 72 and 73: authors should indicate if which people (household, individual or community) seek either early or late treatment

Response: We appreciate the positive feedback from the reviewer .It is corrected and highlighted by track change in the main revised manuscript.

Q: lines 74 to 76: Authors should at least make a case for SSA or Ethiopia as well

Response: More detail information has been provided on these aspects in the introduction section.

Discussion

Q: lines 270 and 271: Authors cannot justify the statement; therefore, it should be rephrased or deleted

Response: We thank the reviewer for the suggestions. we have revised again the document. We have also noted repetitive sentences, unnecessary capitals and smalls, contextual spelling and sentence structure problems. Then we have re-arranged and re-edited the vague sentences. We have provided the document for professional language copy editor/proofreader expert to advanced language edition. Hence clarity problems are resolved and highlighted by red color in the revised manuscript.

Reviewer #2 

Evaluation (review comments for the authors)

This is an important study looking at a significant issue in an understudied context. The authors employ appropriate methods for measuring associations between a variety of factors and treatment-seeking for childhood fever. With some major revisions particularly around the framing of the study (introduction), more information about the variables, sample construction, measurement and analysis (methods), and contextualized results (discussion) this paper can make a contribution to the literature around access and healthcare seeking behavior.

Response: We appreciate the positive feedback from the reviewer.

Q:1) Abstract: cut off – probably due to word count issues. Would like to see a little more summary and contextualization instead of lists of results/AORs – especially because certain terms/concepts haven’t been defined yet when someone is reading abstract so can’t assume reader will be able to understand meaning of results at this point.

Response: thank you for your suggestion. The suggested correction has been made and highlighted in track change.

Q:2)Introduction:

a. First paragraph: Awkward language especially in first sentence. What does the inclusion of evidence from Europe and US add to the study? US numbers not clear 4.4 -7.5% and 30%? Could be because of missing information/typos but it seems like Ethiopia has the highest treatment seeking rates (87.8%) for children of the countries listed so the following sentence about “enhancing treatment-seeking” is less compelling for Ethiopia (although obviously still important for the remaining 12.2% of cases). This could be made clearer and more compelling.

Response: We thank the reviewer for raising this important point. We rewrote the introduction for greater focus. This section has been edited for conciseness by presenting the crucial points of what is known and not known, and why we carried out the study.

Q:b. Overall flow of the introduction could be improved. It also seems to fall short of setting up what the study is really about. The authors provide compelling evidence that fever is linked to Malaria and that early treatment is important for morbidity and mortality, but the introduction is lacking logical flow and clear delineation between content in paragraphs. I would suggest that it be reorganized to set up study, not just in context of rates of fever and why fever matters, but more focus on treatment -seeking behavior.

i. For example, this section should introduce the predictors you seek to measure (generally) and help the reader understand why you are measuring certain individual and community level factors. What are the hypothesized relationships? Is there a conceptual framework or theoretical model that these are based on?

Response: Thank you for your view. This comment is crucial for this manuscript which was our gap. Now we have reviewed works of literature regarding predictors of treatment seeking behaviors in different countries including Ethiopia on the targeted population. Information regarding the issue and the rationale is provided in the introduction section of the revised manuscript and highlighted by track change.

Q:3) Methods and Materials:

a. Section 2.1 should state how child information was collected (e.g. by caregiver report). This comes up again in section 3.2 where it says “of the 10,006 alive children who responded.” I’m assuming that the babies and children are not the ones responding, but this should be very explicit. The total children (10,006) are not mentioned at all in this section. It should be clear where the subsample of 1,354 is drawn from. Also, it seems like caregivers who are not the mother are excluded (grandmothers, aunts, sisters etc.) but it should be explicit that the respondents are the mother to the child for whom the data is reported (not just the caregiver who is a mother).

Response: Thank you for your view. The 2016 EDHS 84915 enumeration areas (EAS) served as a sampling frame, 645 clusters selected in the first stage, and from those clusters 202 were urban and 443 from rural areas. A total of 16,650 households were surveyed in the second stage. The 2016 EDHS interviewed a total 15,683 women between ages of 15–49 year. The data for the present study were extracted as follows: First, women who gave births in the last 5 years were identified. Next, caregivers/mothers who had child with fever in the 2 weeks preceding the survey period were identified. It is customary to get more than one under-five children per a household. However, for data quality, if there were more than one under-five children per household, data were collected from children with the last (recent) birth. Finally, 10,006 were alive children and a total of 1,354 (weighted=1495) mothers/caregivers who had under-five children with fever were included for the analysis. Caregivers were defined as a mother aged between 15-59 years with a child/child under age 5 years, who had responded to the survey. The caregivers of under-five children either mothers or others who are responsible to raise the child and give care was the one who provided the maternal and child related information. Therefore, our study participants were those caregivers who had under five children with febrile illness to assess their health seeking behavior. Therefore, a total of 1354 under five children had history of fever within two weeks. Overall, the study units were the caregivers of under five children with febrile illness and then we went to assess the proportion of mothers who seek health care and the potential individual and community level factors associated with the health seeking behavior of the care givers.

Q:i. What is the distribution of the selected subsample (1,354 children with fever)? We know about EDHS, but does this group have the same characteristics or are they concentrated in certain regional states etc.? This comes up later in discussion section – is this subsample representative? Perhaps as an appendix, it would be helpful to have a comparison of these children and caregiver characteristics compared to the whole sample.

Response: Thank you the reviewer for the comments. As you know some of the regions were oversampled and some of the regions were under sampled, and in addition because of the non-response rate, the EDHS data were weighted for design and non-response to make it nationally representative and to draw valid conclusion. Therefore, for this study we used the weighted data, which was weighted for weighting variable (v005), strata (v021) and primary sampling unit (v023). Besides, considering the hierarchical nature of EDHS data and the child and the caregivers within the cluster/EAs were nested, shares similar characteristics, it could violate the standard logistic regression model assumptions such as independence of observation and homogeneity of variance assumptions. Therefore, we used a multilevel binary logistic regression analysis to get reliable standard error to draw valid conclusions. Due to the non-proportional allocation of the sample across the regions, sampling weights were used to ensure the representativeness of the finding. A sample is a group of people who have been selected for a survey. In the EDHS, the sample is designed to represent the national population age 15-49. However, doing so requires a minimum sample size per area. For the 2016 EDHS, the survey sample is representative at the national and regional levels, and for urban and rural areas. The data were weighted using sampling weight during any statistical analysis to adjust for unequal probability of selection due to the sampling design used in DHS data. Hence, the representativeness of the survey results was ensured.

Q:b. Section 2.2 lists “private sector” “market” as a place to seek care. What is meant by this location?

Response: those locations are a place where health care services delivered and pharmaceutical supplies delivered in the Ethiopian context. Generally, they are health care facilities either governmental or private. 

Q:c. Section 2.2 – would suggest adding another section on analysis and separating from the variables/measures

Response: With great thanks the suggested correction has been made and highlighted in track change. 

Q: d. Section 2.2: The Independent variables listed do not align with what is included in the tables (9+ variables in tables not included in this section). Need more information on the variables. Would like to understand more about why certain predictors were included. What are the hypothesized relationships? Do these come from the literature? Are they context specific or more globally applicable factors? How specific indices were constructed (not just what variables were included in them) and are these based on particular measures from a source (WHO? UNICEF? USAID? Article?) Not all of this needs to go in the methods section, some could be in the introduction, but overall, more information is needed about what is being measuring and why.

Response: thank you for your suggestion. The suggested correction has been made and highlighted in track change.

Q:e. Section 2.2: need to include information on the clusters (“communities”) how are they constructed. Why are they used instead of other levels?

Response: Thank you for the comments. For this data, we fitted a multilevel binary logistic regression analysis by considering factors at individual and cluster/community levels. We considered cluster/EAs as a random variable because at the beginning we consider EA/cluster and region as a random variable and fitted three-level binary logistic regression analysis but the regional level variance was not significant. Besides, to have stable estimate in multilevel binary regression analysis we have to have at least 50 clusters but as you know we have only 11 regions. That is why we fitted a two level multilevel logistic regression analysis considering EA as a random variable.

.How were the levels determined? Fever incidents are nested within children who are nested within caregivers, who are nested in families, who are nested in households, who are nested in communities, cities, states, regions, rural /urban etc. How were certain levels either eliminated (e.g. by selecting only 1 fever incident per child, only 1 child per caregiver, only 1 caregiver per household?) or otherwise accounted for? Of particular importance here, given the variables in your model, is children within caregiver and caregivers within a household.

Response: Thank you reviewer for the comments. Here, in the EDHS 2016 one reproductive age women/caregivers per household were selected and the child related characteristics were extracted about the last child/most recent birth. Therefore, we have one child per household as we have used the KR-file for our study and we do not expect that child were nested within the household rather we expected child was nested within cluster/EA and EA was nested within region. Considering this, we have fitted three-level multilevel binary logistic regression analysis model using STATA command “melogit seeking_for_fever || v001 || region:,” but unfortunately the regional level variance was not significant. Therefore, we fitted two level multilevel logistic regression analysis using individual level and cluster level explanatory variables. THE EDHS 2016 sample were taken by two stage stratified sampling in the first stage enumeration areas that were prepared by central statistical agency were selected in all regions after stratification was done into urban and rural residency. Then after in the second stage households were selected by systematic random sampling technique, and only one caregiver were taken in the household, and data were collected from children with the last (recent) birth if there is more than one under-five children per caregiver. 

Response:

Q:g. Why did you use a cutoff of <=.26 for variable selection? This goes along with more information about the specific variables and why they were included, what they tell us about the research question.

Response: sorry for the typological error that we have made. We use a p-value of 0.2 as a cut-off point in bi-variable analysis as a rule of thumb to control cofounder variables. 

Q:h. How are missing data handled?

Response: Thank you for the comments. We excluded study participants who had missing on the outcome variables but concerning the independent variables we included variables which have no missing and by default as the STATA software is robust it excluded the participants which have missing as it can do complete case analysis. As much as possible we used predictors which did not have missing based on the DHS recode statistics guideline. Missed data in the outcome variables were dropped/excluded, while missing data in the independent variables were managed according to DHS statistics guide (guide to DHS statistics DHS-7 version 2). 

Q:4) Results

Q:a. 3.1 Background Characteristics: Major typo in first sentence that has serious implications if not fixed – “1006 alive children “Further reading showed this should be “10,006” . Another typo in that it references “unmarried as 1,479 (98.98%)” per table this is “married.” Also a minor point on language – “majority” typically means >50% so wouldn’t be used as it is in this paragraph to refer to 27.4% and 26.31%.

Response: thank you for your view. The suggested correction has been made and highlighted in track change. .

Q:b. Table 1: Would be good to include totals so a reader can quickly see the sample size/total frequencies (and assess if they are consistent). As stated before – more information on the variables is needed. There are several variables here that were never mentioned before appearing in the table. While some are more obvious (sex of child), others are somewhat unclear (sex of household?, had diarrhea, had to cough, had anemia) – how were these operational zed, who is being measured, what is the time frame? Why do they matter? Obviously, not all of this information goes in the table, but some could be included here, whereas other info can be in the introduction and methods sections.

Response: Thank you very much for your crucial comments for our manuscript. The suggested correction has been made. We mean to say that sex of household head. The operational definition, time frame and all others were found in EDHS 2O16 report; therefore, we cited it in the corrected documented. 

Q: c. 3.3 Random effect…: More information about the clusters needs to be included in the methods section before reporting on the results by clusters. It is not clear how the clusters are constructed – what is the definition of “community.” How many communities are there? Is there sufficient power for the number of clusters and units within each cluster? The last sentence of this paragraph states that the Model IV was selected because it had the lowest deviance – but the number reported from Table 2 is different than the actual number listed in Table 2 for that model. Which is it?

Response: sorry for the editing error, we wrote null model result on the place of variable category, therefore, we omit it. All the alignment of table lines and other editing issues were resolved in the revised manuscript. 

d. 3.4 Factors associated…: Why state that variables had a statistically significant association at the level of <.20 when in previous sections (methods) you said you were using a cutoff of <0.05 (which is standard) ? In multiple sections/tables the coughing variable is not clearly explained. The text sometimes states that the children were coughing, sometimes the caregiver, sometimes the caregivers “had to cough febrile children.” This should be clear and I believe in the DHS the variable is measuring whether the children had a cough. Same thing applies to diarrhea and rapid breathing. It should be very clear who had the diarrhea and when (same time as fever?) It is assumed this is the child and simultaneous to the fever, but the text/tables are missing important descriptions or stating it in confusing/inconsistent ways (e.g. line 257 “among caregivers that were having short, rapid breathing was 1.68 times…” line 258 “mothers having diarrhea…” line 259 “caregivers who have no diarrhea”)

Response: We very much appreciate this helpful comment. We are grateful for this comment as it points to an important point of view. This comment is crucial for this manuscript which was our gap. all type error, sentence structure, factor interpretation, language usage were resolved and copyedited in the revised document. We used p value less than 0.2 in the bi-variable analysis for selecting candidate variables for multivariable analysis, while p value 0.05 was used to declare the statistical significance in the final model that is multivariable multilevel binary logistic regression. 

e. Table 2: needs proper formatting such as consistent column width.

Response: The suggested correction has been made.

5)Discussion/limitations/conclusion

a. The Malawi numbers listed are inconsistent with what is in the introduction (49.9% vs. 67.3%) assuming these studies are measuring different things but this should be clear.

Response: The suggested correction has been made and highlighted in the revised document.

b. Discussion of sample size seems to be more about the sample construction, not the size? Unless there were issues with too few observations for the multilevel approach. This should be specifically addressed in methods and limitations section.(size, distribution, representativeness).

Response:

c. Would like to see more discussion about the specific predictors not just how they compare to prior studies, but also what they mean in context. Did the hypothesized relationships/effects play out? How do the different predictors relate to each other (or not).

Response:

d. It is not clear that this can be generalized to all Ethiopian caregivers/mothers (also those are different groups) because we don’t know if this small subsample drawn from a representative sample is representative. Are mothers (or caregivers) of children under 5 who had a fever in the last 2 weeks prior to the survey representative of all mothers? Would like to see more in the limitations sections about any potential issues in the methodological and conceptual approaches.

Response: this study uses nationwide representative samples. Therefore, the findings can be generalized to the country/Ethiopian caregivers. As we mentioned the earlier comments, representativeness or generalizability issue was ensured by weighting the data throughout the analysis. Even if the history of fever were traced in the preceding 2 weeks of the caregivers interview, the data were collected for 5 years preceding 2016 EDHS. 

e. Conclusion the mention of diarrheal disease derail treatment-seeking behavior is confusing and doesn’t seem to be in line with results.f. Expected to see these sections linked back to the introduction more in terms of the focus on malaria, implications for the children etc.

Response: : The suggested correction has been made

g. The contribution of this study (which I believe in) isn’t coming through clearly . Being very precise about what was learned, what it means, and how these results can be used/interpreted is key.

Response: : The suggested correction has been made

6) General: Should be copy-edited for typos, awkward language, grammar issues.

Response: We have consumed more time and energy for re-edition of the vague sentences. Based on this, we have rewritten the whole document in a more understandable manner to resolve language problems. Furthermore, we have modified, added, and changed a lot of things start from the title up to references based on other reviewers in addition to you. We are lucky, since the manuscript is assigned for three peer reviewers, and all of the three reviewers comments and questions are different which helps us to learn a lot and modify the whole document.

---

## [Decision Letter · Decision Letter 1]

21 Oct 2021

PONE-D-21-03591R1Individual and community-level factors of treatment-seeking behavior among caregivers with febrile children in Ethiopia: A Multilevel analysisPLOS ONE

Dear Dr. liyew,

Thank you for submitting your manuscript to PLOS ONE. After careful consideration, we feel that it has merit but does not fully meet PLOS ONE’s publication criteria as it currently stands. Therefore, we invite you to submit a revised version of the manuscript that addresses the points raised during the review process.

The revised paper is not adequately addressed concerns raised by the reviewers. I am thus giving another chance to the authors to revised the paper.

We look forward to receiving your revised manuscript.

Kind regards,

Enamul Kabir

Academic Editor

PLOS ONE

Reviewers' comments:

Reviewer's Responses to Questions

**Comments to the Author**

1. If the authors have adequately addressed your comments raised in a previous round of review and you feel that this manuscript is now acceptable for publication, you may indicate that here to bypass the “Comments to the Author” section, enter your conflict of interest statement in the “Confidential to Editor” section, and submit your "Accept" recommendation.

Reviewer #1: All comments have been addressed

Reviewer #2: (No Response)

2. Is the manuscript technically sound, and do the data support the conclusions?

Reviewer #1: Yes

Reviewer #2: Yes

3. Has the statistical analysis been performed appropriately and rigorously? 

Reviewer #1: Yes

Reviewer #2: Yes

4. Have the authors made all data underlying the findings in their manuscript fully available?

Reviewer #1: Yes

Reviewer #2: Yes

5. Is the manuscript presented in an intelligible fashion and written in standard English?

Reviewer #1: Yes

Reviewer #2: (No Response)

6. Review Comments to the Author

Reviewer #1: (No Response)

Reviewer #2: The authors have addressed many of the reviewer comments and the revised manuscript is greatly improved. The introduction does a better job at setting up the paper, the methods section has some additional information, and the results are more clearly presented and contextualized in the discussion. However, there are still some concerns with the paper and several comments from the prior round of review that were not addressed. Comments below are organized by section.

Methods:

1. Several of the prior comments about variables and measures were not addressed. A new section was added and some additional information was provided, but the questions/comments about variable construction, variables used in analysis not mentioned anywhere else until showing in tables, construction of indices etc. were not addressed. Each of the subitems mentioned below was brought up in the prior comments.

a. Additional information about the index variables is needed in the manuscript (see prior comments). Did the community development index and the household wealth index use the same WASH measures? Were all items weighted equally? How exactly were the indices constructed? How were cutoffs and quintiles decided upon? Are there references for these indices? Community level cluster variable based on the WASH and electricity (have other studies used this?)

b. Some issues remain with inconsistencies between what variables are listed in different sections of the paper. For example, the first sentence on independent variables says “size of child at birth” is the measure, but then later in the same paragraph, the authors list under five height for age Z score, weight for age Z score and weight for height Z-score as the measures.

c. No information is given on several measures or why they may be important to include (some of this could go in intro if more appropriate) – examples include but are not limited to mass media exposure, vaccination, place of residence (is this the same as region? ), health insurance (is this yes/no? are there different types of insurance? Do you caregivers need insurance for sick child visits?), How is distance measured (km?, time to get there, ? Does it take main modes of transportation for the study population into account) – Some information can be ascertained from the tables, but the importance of these measures to the study and why they are included (and operationalized in certain ways) is missing for many of them.

d. The authors have not adequately addressed what the “community” level is. Judging by line 187 and line 228, the community is defined as the EDHS EAs, but it would be very helpful to state this explicitly and give the number of clusters etc. in the methods section (beyond the data section that describes how the sampling was done). Also, why is health insurance a community level variable?

e. This is not a major issue – but it is indicative of overlooking specific items brought up by the reviewers: In the author response to a reviewer comment about using a cutoff of p<0.26, they said this was a typo and that it was fixed – line 181 still lists the same cutoff with the typo.

f. The authors responded to a comment about missing data in the response to reviewers, but didn’t mention how they handled missing data in the manuscript – complete case analysis (with % missing)

2. Methods: Because it is such a critical part of the study, I would suggest having a clearer sentence stating your outcome variable – example: “ The outcome variable for this study was treatment-seeking behavior (Yes/ No), defined as whether or not a caregiver sought advice or treatment from a health facility for a living child under-five who had a fever at any time in the 2 weeks preceding the survey.”

Results

1. Why not show your main outcome variable in Table 1?

2. I might also suggest adding a bivariate table. It might be helpful to see some of the bivariate associations/percentages and numbers of observations. You could truncate Table 1 and also not show every variable or every level of every variable in Table 2 if space was a concern.

Editing/Grammar/Formatting:

1. The writing/grammar is much better and really improves readability, however, the manuscript still needs some copyediting. A few notes on this are below.

2. There are some typos and inconsistencies between the different versions of the paper (clean revised manuscript and the tracked changes version). I note some specific inconsistencies below, but this should be thoroughly reviewed. Having two different versions of the revised manuscript made it very difficult to re-review this paper. Which one is the most updated/accurate?

o Introduction: in the clean version, the second sentence of the Intro (lines 55-57) includes a line from the prior sentence (major typo). This seems to be ok in the track changes version. In the clean version there is a sentence on lines 73-73 “In Ethiopia healthcare-seeking behavior is poor….” This sentence does not appear in the tracked changes version

o Paragraph delineation: Examples - in the tracked changes version, on the first page of the introduction there is a new paragraph starting with “World Health Organization.” In the clean version this is lumped in with the prior paragraph (line 68). In the clean version line 81, there is a new sentence starting with “Due to the inadequacy…” but this is an entirely new paragraph in the tracked changes version (also missing a period at the end of the prior sentence).

3. There are different fonts, spacing, sizing used throughout

4. Some capitalization typos – examples include “Children” on lines 134 and 135 (additional typos/grammar issues in the line 134 sentence), Benishangul (line149)

5. The tables need significant formatting work to be more readable – Table 1 is 4.5 pages long (unnecessarily). A reviewer previously suggested adding totals (this does not need to be for every variable – could be overall). Table 2 – one suggestion, you can delete all the lines for the reference group and just note the reference (if not obvious) in first or second column

7. PLOS authors have the option to publish the peer review history of their article (what does this mean?). If published, this will include your full peer review and any attached files.

Reviewer #1: **Yes: **Ebenezer Kwesi Armah-Ansah

Reviewer #2: No

---

## [Author Response · Author response to Decision Letter 1]

3 Dec 2021

Authors’ Response for Reviewers’ Comments

Manuscript ID: PONE-D-21-03591

Title: Individual and community-level factors associated with treatment-seeking behavior among caregivers with febrile children in Ethiopia: A Multilevel analysis

Dear editor(s) and reviewers

First for all we would like to thank you for helping us by reviewing our paper entitled, “Individual and community-level factors associated with treatment-seeking behavior among caregivers with febrile children in Ethiopia: A Multilevel analysis” by giving your precious time. Thus, your comments and questions are constructive and interesting for us. Based on your questions and suggestions we have already modified accordingly.

Reviewers question and/ comments 

Response=> Authors response based on the reviewer(s) questions and comments 

Reviewer #1

Thank you very much for your review. We appreciate the positive feedback from the reviewer

Reviewer #2 

Q1: The authors have addressed many of the reviewer comments and the revised manuscript is greatly improved. The introduction does a better job at setting up the paper, the methods section has some additional information, and the results are more clearly presented and contextualized in the discussion. However, there are still some concerns with the paper and several comments from the prior round of review that were not addressed. Comments below are organized by section.

Response: Thank you very much for your comments and suggestions. Our revised responses based on questions and comments which may help to clarify the paper and its findings are as follows:

Methods:

1. Several of the prior comments about variables and measures were not addressed. A new section was added and some additional information was provided, but the questions/comments about variable construction, variables used in analysis not mentioned anywhere else until showing in tables, construction of indices etc. were not addressed. Each of the subitems mentioned below was brought up in the prior comments.

Response: we had been addressed the comments. 

 a. Additional information about the index variables is needed in the manuscript (see prior comments). Did the community development index and the household wealth index use the same WASH measures? Were all items weighted equally? How exactly were the indices constructed? How were cutoffs and quintiles decided upon? Are there references for these indices? Community level cluster variable based on the WASH and electricity (have other studies used this?)

Response: The wealth index was computed by principal component analysis based on asset based data for urban and rural areas. While the community development index was a composite variable formed from the availability of three basic services in the community: improved water supply, electricity city, and improved sanitation services. It is classified as: Low: community with none of those three services; Medium: community with one or two services; High: community with three services.

b. Some issues remain with inconsistencies between what variables are listed in different sections of the paper. For example, the first sentence on independent variables says “size of child at birth” is the measure, but then later in the same paragraph, the authors list under five height for age Z score, weight for age Z score and weight for height Z-score as the measures.

Response: size of child at birth is to indicate birth weight whereas under five heights for age Z score, weight for age Z score and weight for height Z-score were used as index for assessing the nutritional status during life after birth.

c. No information is given on several measures or why they may be important to include (some of this could go in intro if more appropriate) – examples include but are not limited to mass media exposure, vaccination, place of residence (is this the same as region? ), health insurance (is this yes/no? are there different types of insurance? Do you caregivers need insurance for sick child visits?), How is distance measured (km?, time to get there, ? Does it take main modes of transportation for the study population into account) – Some information can be ascertained from the tables, but the importance of these measures to the study and why they are included (and operationalized in certain ways) is missing for many of them.

Response: the health insurance variable is yes/no type variable and it is community health insurance. In addition, the caregivers might need health insurance because some of the services like lab investigation and medication might not fully available to the public facility. In this case the might utilize the service in private and secured their budget from their insurance. This study utilizes the DHS data from Ethiopia which measures the distance to health facility and categorized as big problem and not a big problem.

d. The authors have not adequately addressed what the “community” level is. Judging by line 187 and line 228, the community is defined as the EDHS EAs, but it would be very helpful to state this explicitly and give the number of clusters etc. in the methods section (beyond the data section that describes how the sampling was done). Also, why is health insurance a community level variable?

Response: community in our case is to mean a group of individuals that shared similar characteristics. As you said the community in EDHS data is EAs and also we had mentioned it on the manuscript its total size and the urban and rural EAs which was 645, 202, and 443 EAs respectively. We used the health insurance variable as community level because the type of health insurance that were collected by the EDHS data was community based health insurance.A community was defined as a group of households sharing a common primary sampling unit/cluster within the dataset. Community level were constructed by aggregating them from individual-level factors. 

e. This is not a major issue – but it is indicative of overlooking specific items brought up by the reviewers: In the author response to a reviewer comment about using a cutoff of p<0.26, they said this was a typo and that it was fixed – line 181 still lists the same cutoff with the typo.

Response: sorry for reaped typological error but we have edited on the revised document.

f. The authors responded to a comment about missing data in the response to reviewers, but didn’t mention how they handled missing data in the manuscript – complete case analysis (with % missing)

Response: as we said in the previous comment, the missing data’s were handled by complete case analysis after checking the% of missing because the variables with missed data had missing value < 5%.

2. Methods: Because it is such a critical part of the study, I would suggest having a clearer sentence stating your outcome variable – example: “ The outcome variable for this study was treatment-seeking behavior (Yes/ No), defined as whether or not a caregiver sought advice or treatment from a health facility for a living child under-five who had a fever at any time in the 2 weeks preceding the survey.”

Response: thank you for your constructive comment, we accept it and change by your ideas

Results

1. Why not show your main outcome variable in Table 1?

Response: We thank the reviewer for the suggestions. We have included the suggested changes and track changed in the revised manuscript.

2. I might also suggest adding a bivariate table. It might be helpful to see some of the bivariate associations/percentages and numbers of observations. You could truncate Table 1 and also not show every variable or every level of every variable in Table 2 if space was a concern.

Response: We appreciate the positive feedback from the reviewer, based on your comment we have modified this section in understandable and precise way.

Editing/Grammar/Formatting:

1. The writing/grammar is much better and really improves readability, however, the manuscript still needs some copyediting. A few notes on this are below.

2. There are some typos and inconsistencies between the different versions of the paper (clean revised manuscript and the tracked changes version). I note some specific inconsistencies below, but this should be thoroughly reviewed. Having two different versions of the revised manuscript made it very difficult to re-review this paper. Which one is the most updated/accurate?

Response: After reviewer(s) suggestions, we have revised again the document and we also have found grammatical, contextual unclarity and sentence structure problems. Then we have consumed more time and energy for re-edition of the vague sentences. Based on this, we have rewritten the whole document in more understandable to resolve language problems. 

Introduction: in the clean version, the second sentence of the Intro (lines 55-57) includes a line from the prior sentence (major typo). This seems to be ok in the track changes version. In the clean version there is a sentence on lines 73-73 “In Ethiopia healthcare-seeking behavior is poor….” This sentence does not appear in the tracked changes version

Response: Thank you for your comments, the suggested correction has been made. 

o Paragraph delineation: Examples - in the tracked changes version, on the first page of the introduction there is a new paragraph starting with “World Health Organization.” In the clean version this is lumped in with the prior paragraph (line 68). In the clean version line 81, there is a new sentence starting with “Due to the inadequacy…” but this is an entirely new paragraph in the tracked changes version (also missing a period at the end of the prior sentence).

 Response: It has been rewritten again by incorporating necessary information in this regard.

3. There are different fonts, spacing, sizing used throughout

Response: The font size and type has been checked and it is consistent in the whole document. The used font type and size is “Times New Roman” by “12”. 

4. Some capitalization typos – examples include “Children” on lines 134 and 135 (additional typos/grammar issues in the line 134 sentence), Benishangul (line149)

Response: the correction has been made. 

5. The tables need significant formatting work to be more readable – Table 1 is 4.5 pages long (unnecessarily). A reviewer previously suggested adding totals (this does not need to be for every variable – could be overall). Table 2 – one suggestion, you can delete all the lines for the 

Response: the correction has been made.

Finally, we thank the reviewers’ editors for their kind comments, constructive criticisms and useful suggestions for which we have used to improve our manuscript. We have re-read, edit and rewrite the whole manuscript once again and made any necessary editorial corrections. Thank you for the helpful review once again.

---

## [Decision Letter · Decision Letter 2]

16 Feb 2022

Individual and community-level factors of treatment-seeking behavior among caregivers with febrile children in Ethiopia: A Multilevel analysis

PONE-D-21-03591R2

Dear Dr. liyew,

We’re pleased to inform you that your manuscript has been judged scientifically suitable for publication and will be formally accepted for publication once it meets all outstanding technical requirements.

Kind regards,

Enamul Kabir

Academic Editor

PLOS ONE

Additional Editor Comments (optional):

Reviewers' comments:

Reviewer's Responses to Questions

**Comments to the Author**

1. If the authors have adequately addressed your comments raised in a previous round of review and you feel that this manuscript is now acceptable for publication, you may indicate that here to bypass the “Comments to the Author” section, enter your conflict of interest statement in the “Confidential to Editor” section, and submit your "Accept" recommendation.

Reviewer #1: All comments have been addressed

Reviewer #2: (No Response)

2. Is the manuscript technically sound, and do the data support the conclusions?

Reviewer #1: Yes

Reviewer #2: Yes

3. Has the statistical analysis been performed appropriately and rigorously? 

Reviewer #1: Yes

Reviewer #2: Yes

4. Have the authors made all data underlying the findings in their manuscript fully available?

Reviewer #1: Yes

Reviewer #2: Yes

5. Is the manuscript presented in an intelligible fashion and written in standard English?

Reviewer #1: Yes

Reviewer #2: Yes

6. Review Comments to the Author

Reviewer #1: Discussion & Conclusion:

How do you explain the findings? What are the implications? The discussion could be strengthened by linking the results to previous studies. I think the authors can provide more specific policy implications based on these interesting findings.

Please kindly discuss this matter.

Reviewer #2: The authors have addressed the reviewer comments and the manuscript is acceptable for publication. I'm only including a few minor formatting/copy editing suggestions here to be helpful to the authors in their final reviews, but these do not impact the overall content of the manuscript.

1. Spacing: Spaces between text and citations – sometimes there is a space, sometimes there is not between the text and the citation. We suggest going through and adding spaces consistently throughout the manuscript. Similarly, there is inconsistent spacing in the results when they are listed in parentheses. Sometimes there is a space and sometimes not. Suggest adding spaces throughout (examples in lines 235-247). This issue is also present with the symbol % - sometimes a space, sometimes not. Suggest no spaces between the number and % (e.g. 54.0%, but add a space after the % symbol, e.g. 95% CI instead of 95%CI). Examples lines 249-270.

2. Line 190 has a minor typo. Should be a comma not a period: “To accommodate for the complex sampling design employed in the survey, weighted data analysis was employed.”

3. Starting on line 196: sometimes the authors capitalize “Model” when referring to a specific model (e.g. Model III), sometimes they do not “model II.”

4. Tables 1 &2: Capitalization in tables is inconsistent (in respondent’s characteristic column). We also suggest adding an “s” and making characteristics plural. Justification in the columns is also inconsistent (some are centered some are left).

5. Line 235 – typos in the OR, has a comma instead of a period for “38, 86” instead of “38.86”

7. PLOS authors have the option to publish the peer review history of their article (what does this mean?). If published, this will include your full peer review and any attached files.

Reviewer #1: No

Reviewer #2: No

---

## [Editor Report · Acceptance letter]

8 Mar 2022

PONE-D-21-03591R2 

Individual and community-level factors of treatment-seeking behaviour among caregivers with febrile children in Ethiopia: A Multilevel analysis 

Dear Dr. Liyew:

I'm pleased to inform you that your manuscript has been deemed suitable for publication in PLOS ONE. Congratulations! Your manuscript is now with our production department. 

Kind regards, 

on behalf of

Dr. Enamul Kabir 

Academic Editor

PLOS ONE